# Mechanisms of Disulfide Bond Formation in Nascent Polypeptides Entering the Secretory Pathway

**DOI:** 10.3390/cells9091994

**Published:** 2020-08-29

**Authors:** Philip J. Robinson, Neil J. Bulleid

**Affiliations:** Institute of Molecular, Cell and Systems Biology, College of Medical Veterinary and Life Sciences, Davidson Building, University of Glasgow, Glasgow G12 8QQ, UK; philip.robinson@glasgow.ac.uk

**Keywords:** disulfide formation, protein folding, protein secretion, protein synthesis, PDI, ER

## Abstract

Disulfide bonds are an abundant feature of proteins across all domains of life that are important for structure, stability, and function. In eukaryotic cells, a major site of disulfide bond formation is the endoplasmic reticulum (ER). How cysteines correctly pair during polypeptide folding to form the native disulfide bond pattern is a complex problem that is not fully understood. In this paper, the evidence for different folding mechanisms involved in ER-localised disulfide bond formation is reviewed with emphasis on events that occur during ER entry. Disulfide formation in nascent polypeptides is discussed with focus on (i) its mechanistic relationship with conformational folding, (ii) evidence for its occurrence at the co-translational stage during ER entry, and (iii) the role of protein disulfide isomerase (PDI) family members. This review highlights the complex array of cellular processes that influence disulfide bond formation and identifies key questions that need to be addressed to further understand this fundamental process.

## 1. Introduction

Cell compartmentalisation during the evolution of eukaryotic cells resulted in membrane-bound organelles with specialised functions. The endoplasmic reticulum (ER) is an organelle involved in the sorting and targeting of newly synthesised proteins. It is estimated that approximately 30–40% of genes encode for proteins that are targeted to the ER [1,2], most of which enter co-translationally via the sec translocon. As the entry point to the secretory pathway, the ER lumen is the point at which nascent secretory proteins fold into their native three-dimensional (3D) structures. Approximately 80% of secretory proteins contain disulfide bonds [3], the majority of which form during the folding process.

Disulfide bond formation in proteins occurs exclusively between cysteine sidechains via the oxidation of thiol groups (Figure 1A). The cytosol is an unfavourable environment for disulfide formation because it contains robust NADPH-dependent reducing pathways to maintain proteins in a reduced form [4]. In contrast, the ER contains distinct enzymatic pathways involving the protein disulfide isomerase (PDI) family of proteins that channel oxidising equivalents to specific substrates [5,6]. Reductive pathways are also present in the ER, which are required to correct non-native disulfide bonds during folding or to reduce disulfide bonds before degradation [7]. Hence a balanced redox poise is required in the ER to enable disulfide formation, reduction, and isomerisation to take place in nascent proteins. Maintenance of this redox balance is considered to be aided by millimolar concentrations of glutathione [8]. This ubiquitous tripeptide molecule, composed of cysteine, glycine, and glutamic acid, can exist in both oxidised (GSSG) and reduced forms (GSH). In the cytosol, the activity of glutathione reductase maintains a high ratio of GSH to GSSG [9]. Glutathione reductase is absent in the ER and, instead, the presence of oxidative pathways lowers the GSH-to-GSSG ratio [10]. The presence of this pool of glutathione is likely to contribute to mechanisms that counter oxidative and reductive stress in the ER.

The mechanisms of protein folding were mainly studied through in vitro unfolding and refolding experiments using full-length proteins [11,12]. To extrapolate these findings to an ER setting, we must consider the vectorial nature of protein synthesis, ER translocation, and the specialised folding conditions of the ER. In this review, we discuss the mechanisms of disulfide formation during folding in this context. Firstly, we discuss the biophysics of disulfide bonding in relation to folding and how this correlates with proposed folding mechanisms. Then, we highlight how these mechanisms apply to nascent polypeptides as they emerge into the ER lumen. Finally, we look at how and when ER resident factors interact with nascent polypeptides undergoing folding.

## 2. Mechanisms of Disulfide Bond Formation during Protein Folding

Cysteine residues in proteins are frequently found in the interior of the protein fold, where they are sterically hindered from undergoing disulfide bond formation, reduction, or rearrangement. This constraint means that disulfide formation is more likely to occur at earlier stages in the folding process. The relationship between folding and disulfide formation was first experimentally demonstrated almost 50 years ago by the refolding of ribonuclease [13]. This experiment showed that refolding from a reduced, denatured precursor results in a single native disulfide pattern, despite over 100 disulfide bond configurations being possible. Hence, the correct disulfide bond pattern must be encoded in the amino-acid sequence and must form during the folding process. Since then, many in vitro refolding experiments showed that disulfide bonds are required to enable specific proteins to refold [14]. This led to a consensus that disulfide bonds form first before folding and subsequently drive the formation of structure [15]. Recent studies showed that this is not always the case, and disulfide bonding can also follow nascent structure formation [16,17,18].

To understand the mechanisms of disulfide formation during folding, it is important to consider the physical requirements for cysteine coupling. Paired cysteines are often separated in the amino-acid sequence and must come into close proximity and in the correct orientation for the disulfide bond to form [5,19] (Figure 1A). The two thiol groups cannot react spontaneously, but require small-molecule oxidants or enzymes to catalyse the process [2]. The surrounding microenvironment also influences the pKa of the cysteine-thiol group and, therefore, its ability to deprotonate and form the significantly more reactive thiolate species [20]. How do these requirements fit with the conformational folding processes that occur during structure formation? In Figure 1B, we outline a mechanistic scheme to describe the potential folding pathways that disulfide containing proteins can follow.

At the pre-folding stage, the polypeptide exists as a random coil, which is dynamic and can potentially undergo disulfide formation through internal random collisions. Here, disulfide bond formation is not truly random because it is influenced by the non-covalent interaction of local sidechains and the positioning of cysteine residues in the amino-acid sequence [21]. The resulting disulfides can introduce stability to the molecule and drive further folding or introduce structural strain to the molecule [22], and be prone to reduction and rearrangement. Following the pre-folding stage, folding coupled disulfide formation takes place; this stage broadly spans the formation of the nascent tertiary structure and native disulfides, before folding is complete. The timing of structure formation relative to disulfide formation during folding-coupled disulfide formation can be described by the two mechanisms outlined below [23].

In the structured precursor mechanism (Figure 1B red arrows), the transition from an unstructured precursor to a structured, native-like intermediate occurs before native disulfide formation [23]. The presence of the nascent tertiary structure positions cysteine side chains, increasing the localised concentration of the thiol groups, to favour specific cysteine coupling, while remaining accessible to catalysing factors. Nascent tertiary structure can also increase the reactivity of cysteine sidechains by promoting interactions that stabilise the thiolate ion [20]. Once formed, the presence of the disulfide bond covalently locks the structure in place, adding stability to the final fold.

In the quasi-stochastic mechanism (Figure 1B blue arrows), disulfide bonding occurs before conformational folding. Cysteine sidechains pair in the unstructured precursor and stabilise the polypeptide by restricting its conformational dynamics [24]. Once formed, the disulfides provide constraints for further folding and can drive the formation of the tertiary structure. The quasi-stochastic mechanism is favoured if the structure of the folded protein is largely defined by its disulfide bonding or if disulfide bonds make up the hydrophobic core of the protein.

Most disulfide-containing proteins follow a folding mechanism that is a hybrid of the models described above. Individual domains and structures may favour different characteristics of these models in order to achieve the native fold. Another factor to consider is the formation of non-native disulfide bonds. These can be unwanted misfolded species or productive intermediates that are necessary to form the native structure. Two extreme models that describe folding mechanisms either in the absence or in the presence of non-native disulfide bonds are represented by the proteins bovine pancreatic trypsin inhibitor (BPTI) and hirudin. If native disulfide bonds are the only disulfides that form during folding, then the mechanism is described as “BPTI-like” [25]. This is favoured by but not exclusive to the structured precursor mechanism of folding. The alternative “hirudin-like” model is characterised by heterogeneous folding intermediates and the presence and rearrangement of non-native disulfide bonds. These non-native species are more likely to form at the pre-folding stage and are favoured by the quasi-stochastic mechanism of folding.

## 3. Disulfide Bond Formation in Nascent Polypeptides during Translocation

### 3.1. The Folding of Proteins at the Co-Translational Stage

The above mechanisms describe the folding of proteins from unstructured, full-length polypeptides into their native states, as they apply in an in vitro system. To assess how these mechanisms apply to the folding of proteins in cells, the vectorial nature of synthesis must also be taken into account. The translation speed of 1–5 amino acids per second [26] means that secretory proteins gradually enter the ER lumen, providing time for folding to begin at the N-terminus while translation continues. This process of co-translational folding has some major physical constraints in comparison to folding of released proteins (Figure 2). The tethering of the C-terminus to the P-site of the ribosome restricts movement and sequesters a substantial amount of the polypeptide in the ribosome–sec complex. With only partial exposure of the N-terminal amino-acid sequence to the ER, the intrachain interactions and subsequent folding processes are limited. During translocation, the exposed nascent chain is also localised in close proximity to the membrane [27], favouring interactions with membrane-bound factors, including enzymes responsible for glycosylation and signal peptide cleavage [28]. On release from the ribosome, soluble proteins can diffuse further into the ER lumen, providing greater conformational space for folding. The C- and N-termini are now able to come into close contact, which is often an important nucleation event for further folding [29,30]. Many proteins are membrane-anchored, and release allows them to laterally diffuse in the lipid bilayer. ER factors such as chaperones, which can bind at the co-translational stage, can also interact following ribosomal release. These factors protect and process nascent proteins and restrict their progression through the secretory pathway until folding is complete [31].

The relevance of co-translational folding is perhaps best exemplified by mutations that result in a change in codon without an accompanying change in amino acid. These synonymous mutations may influence the rate of amino-acid incorporation, due to the variance in the abundance of transfer RNA (tRNA). This can alter the time available for folding intermediates to accumulate, which can lead to alternate folding outcomes [26,32]. Despite the importance of co-translational folding, our knowledge of this process is restricted by the dynamic nature of nascent chains, which makes them challenging to study with high-resolution structural techniques [33]. The complexity of the system also hinders analysis with biophysical techniques, which are typically used to study the folding of purified proteins. Co-translational folding data were obtained in prokaryotic systems through NMR [34,35], but the amount of material required currently hinders studies in eukaryotic systems. Instead, our knowledge of these processes was mainly obtained through biochemical assays that can measure indicators of folding such as catalytic activity [36], disulfide formation [37], and protease susceptibility [16]. Such studies characterised the timing and mechanism of folding relative to translocation in specific substrates, capturing a broad range of folding events from secondary structure formation, which occurs before nascent chains leave the ribosome exit tunnel [38], to the assembly of large multichain complexes [39].

### 3.2. Capturing Disulfide Bond Formation during Folding in Cells

The detection of disulfide-bonded species during folding in cells is greatly aided by using cell-permeable alkylating agents such as *N*-ethylmaleimide (NEM). These reagents irreversibly block free thiol groups, allowing for samples to be processed for analysis without further changes in disulfide bonding. Acid quenching can also be used for the same purpose, by lowering the pH to prevent deprotonation of the cysteine thiol group, thereby prohibiting formation of the reactive thiolate ion that participates in thiol–disulfide exchange reactions [40]. Following blocking and processing, disulfide bonds can be detected by comparing reduced and non-reduced samples using separation techniques such as SDS-PAGE. These methods take advantage of changes in mobility caused by disulfide bonds, typically resulting in faster migration in comparison to reduced counterparts [2]. Through these methods, transient species can be captured and detected, allowing for changes in disulfide bonding to be assessed over time. To also understand how disulfide formation correlates with folding requires methods to define the formation of native structure. This can be achieved using direct assays, such as immunoisolation to detect native epitopes [41] or proteolysis to distinguish between folded and unfolded states [16]. Indirect assays can also confirm correct folding; these include the assembly of substrates into oligomeric complexes or the *O*-linked glycosylation of substrates that progress through the secretory pathway [37].

### 3.3. Evidence for Disulfide Bond Formation in Translation Intermediates

The first evidence for co-translational disulfide formation during ER entry was published over 40 years ago when disulfide bonds were detected in substrates undergoing translocation [42,43]. Since then, assays were developed to study disulfide formation using large, multidomain substrates which are long enough to allow for a substantial proportion of the N-terminus to be ER-exposed before release (Figure 3A). For example, the type I membrane glycoprotein haemagglutinin (HA) (Figure 3B) was studied with pulse-chase radiolabelling assays that are able to detect disulfide bonds in species undergoing translocation [44,45]. It was shown that specific disulfide bonds form at the co-translational stage, which is indicative of directed cysteine coupling. These disulfides were not present in all nascent chains, revealing heterogeneity to the process [45]. The long-range disulfide bonds form exclusively after release as part of a defined folding pathway containing structured intermediates [44,46]. In other studies, the low-density lipoprotein receptor (LDLR) was used as a substrate. In this case, long-range, non-native disulfide bonds were identified at the co-translational stage that are productive intermediates on the folding pathway [37]. Recent evidence shows that rearrangement of these non-native disulfide bonds begins at the co-translational stage once downstream sequences are exposed [47]. This challenges the assumption that native folding occurs in a modular fashion from the N- to C-terminus.

Further studies with HA used dithiothreitol (DTT), a cell-permeable reducing agent that inhibits disulfide formation without disrupting, translation, translocation, and other ER processes [41]. Treatment with DTT followed by its subsequent removal allows for disulfide formation to be induced at the post-translational stage. Using this method, it was shown that disulfide bonds can form correctly post-translationally in HA, leading to native folding of the protein [41]. However, in a separate study using the related protein hemagglutinin-neuraminidase (HN), native folding was prevented if the disulfide bonds form post-translationally [49]. This illustrates how co-translational disulfide formation can be crucial for correct folding in specific substrates.

An alternative approach to using large substrates for the study of co-translational disulfide formation is to add a C-terminal extension to a smaller domain. This extension acts as a linker that allows ER exposure of the N-terminal domain, while the C-terminus remains attached to the ribosome (Figure 3C). This strategy was recently used in a eukaryotic translation system to monitor disulfide formation, in different folding domains, relative to translocation [16,50]. For this purpose, stalled translation intermediates of different lengths were produced that represent different stages of translocation. In this system, templates are designed to control the length of translation intermediates and, consequently, the degree of ER-exposure. This level of control enables a detailed analysis of how ER exposure at the N-terminus of the nascent polypeptide correlates with its folding. In these experiments, three proteins were compared with different structures and disulfide densities: β2M (single disulfide, β-sheet rich), prolactin (three disulfides, α-helix rich), and the disintegrin domain of ADAM10 (seven disulfides that define the structure) (Figure 3D). For the extended-β2M substrate, disulfide formation occurred in translation intermediates but only following the formation of nascent structure and full exposure of the domain [16]. For extended-prolactin, the long-range disulfide was absent at all intermediate lengths tested until release was initiated [50]. The ADAM10 disintegrin domain was the only protein which formed disulfides when partially exposed to the ER lumen [50]. This resulted in heterogeneous disulfide-bonded isomers and the presence of non-native disulfides, which were rearranged once reducing pathways were activated. From these results, the mechanisms of disulfide formation, as described in Figure 1B, can be inferred for the three proteins. For β2M and prolactin, disulfide formation requires full domain exposure and folding, which fits with the structured precursor mechanism of folding. In contrast, the disintegrin domain follows the quasi-stochastic mechanism of folding, as disulfides form stochastically much earlier in the translocation process before the structure is defined.

## 4. The Role of PDI Family Members in Nascent Polypeptide Folding

### 4.1. The PDI Family Acts via the Mechanism of Thiol–Disulfide Exchange

In the above sections, we discussed the mechanisms of disulfide formation during folding and how they take place at the co-translational stage, but we have yet to look at the essential role of catalysts. Protein disulfide isomerase (PDI) is the archetypal member of a family of a proteins, which are the primary catalysts for disulfide bond formation, reduction, and isomerisation in the ER [51,52,53,54]. The PDI family members are structurally characterised by thioredoxin folds [55] with catalytic centres containing CXXC motifs [56,57] (Figure 4A). In order to catalyse disulfide formation, PDI family members act through the mechanism of thiol–disulfide exchange, a process via which they donate a disulfide bond to a substrate and, in turn, become reduced (Figure 4B). The first step in this reaction is nucleophilic attack by a thiolate ion from the substrate onto the active site disulfide of the PDI, to produce a mixed disulfide intermediate (Figure 4B step 1). This is later followed by a second nucleophilic substitution to release reduced PDI and the disulfide-bonded substrate (Figure 4B step 3) [2]. This mechanism involves two S_N_2 reactions that require a linear trisulphur transition state [58] (Figure 4C) and, thus, the active site of the disulfide donor must be aligned with the cysteine residues of the reduced acceptor to satisfy this steric requirement. Via the reverse reaction, certain PDI family members can reduce disulfide bonds [59], a process that facilitates protein degradation and enables non-native disulfides to be rearranged during folding [7]. Whether a PDI acts as a reductant or oxidant largely depends on the reduction potential of the active site, which is influenced by the two amino acids between the cysteines [60] and the surrounding microenvironment [53]. The thiol–disulfide exchange process is also dependent on pH, as this affects the deprotonation of the cysteine thiol group to form the reactive thiolate ion, which is required for nucleophilic attack to take place.

### 4.2. Oxidation Pathways to Re-Activate PDI Family Members Following Catalysis of Disulfide Bond Formation

By oxidising substrates during disulfide bond formation, PDI family members themselves become reduced and require subsequent re-oxidation in order to participate in further cycles of disulfide exchange. For this purpose, enzymatic pathways are present in the ER that introduce disulfide bonds into the active sites of PDI family members [61]. A major pathway involves the flavoenzyme ER oxidoreductin 1 (Ero1). This enzyme transfers electrons from reduced PDI, via an FAD (Flavin adenine dinucleotide) cofactor to molecular oxygen, generating oxidised PDI and hydrogen peroxide [62]. Hydrogen peroxide is a reactive oxygen species and, thus, the activity of Ero1 can cause oxidative stress. In addition to hydrogen peroxide removal, the activity of Ero1 must be tightly regulated, and feedback mechanisms exist to inhibit its activity in the event of ER stress [63,64]. Other enzymes that oxidise PDI family members include peroxiredoxin IV [65] and the glutathione peroxidases, Gpx7 and Gpx8 [66], all of which generate disulfide bonds by reducing hydrogen peroxide to water, thereby also contributing to hydrogen peroxide detoxification. Another pathway to generate disulfide bonds involves the vitamin K epoxide reductase (VKOR) [67], this is a transmembrane protein that can oxidise thioredoxin-like proteins as part of its catalytic cycle, in which vitamin K epoxides become reduced. In addition to these enzymatic pathways, small molecules such as hydrogen peroxide and glutathione can also oxidise PDIs directly [5]. Once PDIs are oxidised, they can transfer disulfide bonds to reduced PDIs through thiol–disulfide exchange, providing a route for disulfide bonds to be shuttled from de novo sources throughout the network of PDI family members [68].

### 4.3. PDI Preferentially Interacts with Unfolded Substrates to Catalyse Disulfide Bond Formation

PDI family members act on a broad range of substrates and, therefore, must have a promiscuous mechanism of binding. However, the mode of interaction must still be precise to position cysteines and satisfy the steric requirements for thiol–disulfide exchange to occur [5]. Molecular dynamics simulations and single-molecule studies showed that the catalytic domains of PDI are flexible, which helps to achieve such interactions in structurally diverse substrates [69,70]. When considering the mode of PDI–substrate interaction, it is also important to take into account the accessibility of cysteines in the substrate. In fully folded proteins, disulfide bonds are often internalised and inaccessible. In order to catalyse disulfide formation, PDI must gain access to cysteine residues at an early stage of folding. Evidence from in vitro binding studies showed that PDI interacts with the exposed hydrophobic regions of polypeptides, which indicates a high affinity for unfolded or partially folded substrates, and a decreasing affinity for fully folded substrates [71]. Considering this, it is likely that PDI family members bind to nascent polypeptides as they are translocated into the ER. Such interactions were captured for specific substrates using crosslinking assays [16,72]. This also fits with the evidence that disulfide formation takes place at the co-translational stage, as described in Section 3.

### 4.4. Evidence for Disulfide Reduction and Rearrangements at the Co-Translational Stage

Achieving a native disulfide pattern during folding, for proteins that contain more than two cysteine residues, often involves the reduction and rearrangement of non-native disulfide bonds. These must be surface-exposed in order to be accessible to attacking thiols during isomerisation [73,74]. It is well established that reducing PDIs act on released substrates in the ER; however, can they also act on substrates at the co-translational stage? Disulfide rearrangements require reducing power, which originates in the cytosol [75] and transfers across the membrane via an unidentified transmembrane protein [76], for use by reducing PDIs such as ERdj5 [77] and ERp57 [78]. By controlling the activity of these pathways, disulfide rearrangements can be monitored in translation systems [50,75]. These methods were used to show that disulfide rearrangements can take place in stalled translation intermediates [50], indirectly revealing that reducing PDIs can act before release is initiated. Other studies showed that ERp57 crosslinks to HA at the co-translational stage [79], where it is likely to reduce non-native disulfide bonds that form during translocation. A recent study using LDLR as a substrate also showed that rearrangement of disulfide bonds occurs while the protein is undergoing translocation [47]. Collectively, these studies provided evidence that the co-translational stage of ER entry is not only a time point for the formation of disulfide bonds, but also the subsequent reduction and rearrangement of non-native disulfide bonds.

## 5. Conclusions and Future Directions

Disulfide bond formation is a fundamental biochemical process that takes place in structurally diverse folding domains across the proteome. Research into disulfide formation directly benefits human health, by providing molecular insight into disease origins, such as those that involve protein misfolding or oxidative stress. It can also help to enhance the production of biopharmaceuticals, such as hormones and antibodies, to improve efficiency and lower production costs. In this paper, the mechanisms via which disulfides form during folding in cells was reviewed with focus on events that take place during translocation. At this stage of synthesis, the translocation machinery imposes physical constraints on the nascent polypeptide. This makes co-translational folding significantly different to the folding of full-length, released proteins and technically more difficult to study. Overall, the aim of this review was to provide insight into the complex processes that occur during folding and disulfide formation at the co-translational stage. Despite huge advances in our understanding of this topic, key questions remain unanswered, some of which are outlined below.

### 5.1. What Are the Structures of Translation Intermediates and How Do They Change throughout Translocation?

Recent advances in NMR and cryo-EM enabled translation intermediates to be studied at high resolution [33], but the complexity and dynamics of the translation systems, and the transient nature of folding intermediates make this process both limited and challenging. Ultimately, an approach combining data from both real-time biochemical assays and high-resolution techniques is required to truly understand the sequence of events that occur at the molecular level.

### 5.2. How Does Our Understanding of Co-Translational Folding and Disulfide Bond Formation Apply across the Proteome?

To date, only a small number of select substrates were studied in detail, which gives a limited view of the folding mechanisms involved at the co-translational stage. More substrates need to be studied and global approaches need to be developed to understand the rules of folding in greater depth.

### 5.3. How and When Do Folding Factors Interact with Nascent Polypeptides?

The ER contains a diverse network of folding factors, and it is difficult to define how and when they interact with specific substrates. One possible approach to address this is ribosome profiling, which measures the distribution of ribosomes across messenger RNA (mRNA) molecules during translation [80]. Combining this technique with immunoisolation strategies can identify how chaperone interactions with nascent polypeptides correlate with translation and translocation. The wealth of information provided by such analysis will improve understanding of how the ER–chaperone networks operate on folding substrates in the cell. How post-translational modifications such as glycosylation [81] and signal peptide cleavage [82] both influence and co-ordinate with disulfide formation, folding, and chaperone interactions is also a highly relevant question that requires further study.

Overall, protein folding in the ER is influenced by many factors, each of which require further exploration in order to elucidate how native structures form efficiently. Our understanding of disulfide bond formation in the cell advanced dramatically over the past 50 years, but there are still plenty of questions left to answer, in order to determine the precise mechanisms via which correct disulfide bonding is achieved for the many proteins that traverse the secretory pathway.

## Figures and Tables

**Figure 1 cells-09-01994-f001:**
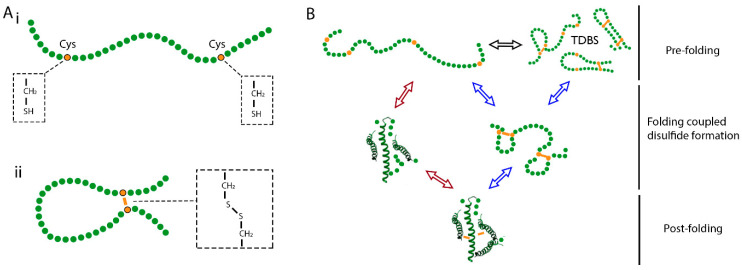
Disulfide bond formation and conformational folding. (**A**) Schematic diagram of a nascent polypeptide, with the molecular structure of cysteine sidechains highlighted before (**i**) and after (**ii**) disulfide bond formation. (**B**) Mechanistic scheme to describe folding coupled disulfide formation. At the pre-folding stage, the nascent polypeptide exists as a dynamic random coil, in which disulfide interchange can take place to form transient disulfide bonded species (TDBS). If folding follows the folded precursor mechanism (red arrows), then formation of the nascent tertiary structure occurs first and promotes native disulfide formation. If folding follows the quasi-stochastic mechanism (blue arrows), then disulfide formation occurs first and drives the formation of the native tertiary structure. At the post-folding stage, the native fold is complete, and disulfide bonds are buried and protected.

**Figure 2 cells-09-01994-f002:**
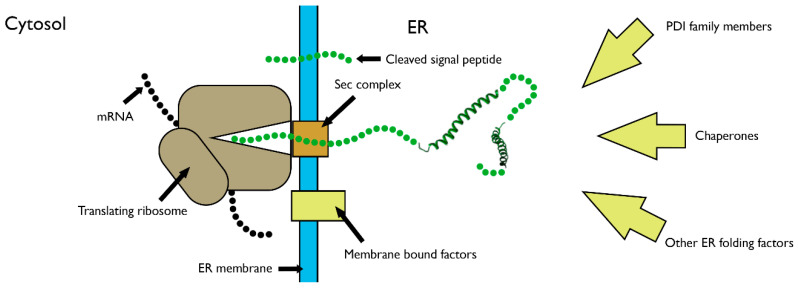
Folding of a ribosome-associated nascent chain during endoplasmic reticulum (ER) entry. Schematic showing a translating ribosome targeted to the ER membrane. The nascent polypeptide (green) enters the ER via the sec complex, undergoes signal peptide cleavage, and begins to fold at the ER-exposed N-terminus. Both membrane-bound and soluble ER factors can interact with the nascent polypeptide as it undergoes translocation.

**Figure 3 cells-09-01994-f003:**
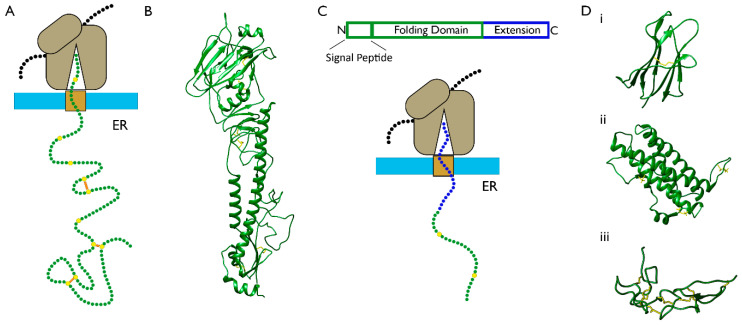
Model proteins to study co-translational disulfide bond formation. (**A**) Schematic showing an ER-targeted, ribosome-associated nascent chain complex in which disulfide formation takes place before translation is complete. (**B**) Ribbon diagram representing the three-dimensional (3D) structure of the haemagglutinin ectodomain (HA) monomer (Protein Data Bank (PDB) 1HA0). (**C**) Extensions to proteins at the C-terminus enable full translocation of the N-terminal domain while the C-terminus remains ribosome-attached. (**D**) Ribbon diagrams representing the 3D structures of (**i**) β2M (PDB 1A1M), (**ii**) prolactin (PDB 1RW5), and (**iii**) the disintegrin domain of ADAM10 (PDB 6BE6). Molecular graphics for figures were performed with UCSF Chimera version 1.14, developed by the Resource for Biocomputing, Visualization, and Informatics at the University of California, San Francisco [48].

**Figure 4 cells-09-01994-f004:**
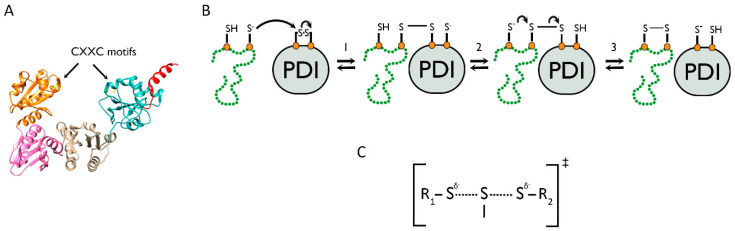
The mechanism of thiol–disulfide exchange in protein disulfide isomerase (PDI)-catalysed disulfide formation. (**A**) Ribbon diagram representing the 3D structure of PDI (PDB 2B5E), which contains four thioredoxin domains. The locations of the CXXC motifs are highlighted. (**B**) The reaction mechanism for thiol–disulfide exchange between a substrate and PDI, in which PDI oxidises the substrate and in turn becomes reduced. (**C**) The linear trisulphur transition state that forms during the S_N_2 reactions at steps 1 and 3 as shown in (**B**).

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
