# Peer review of "Mechanisms of Disulfide Bond Formation in Nascent Polypeptides Entering the Secretory Pathway"

_cells, 2020, doi:10.3390/cells9091994_

Round 1

Reviewer 1 Report

This is a nice piece on oxidative protein folding in the ER with a declared focus on co-translational processes. The text is well written, mostly well cited and the figures nicely line up with it and provide additional clarity. In my opinion, however, although the authors did a good job, a few "gaps" still need to be filled before publication. Thus, to fully "highlight the complex array of cellular processes that influence disulfide formation" (abstract), sections on ER oxidases, glutathione and pH/pKa should be added:

ER oxidases: The one sentence in the introduction on "enzymatic pathways that channel oxidizing equivalents to specific substrates" is not enough. Furthermore, the referencing (current refs 5-7) is unsatisfying: An outdated reference on oxidized glutathione in the secretory pathway, a newer self-citation on thiol-disulfide exchange between PDIs, and an old reference on yeast Ero1. Indeed, the latter was identified before its mammalian orthologs, but if original papers are to be cited, the two papers by the Kaiser and Weissman labs from 1998 would be more appropriate. In my opinion, however, yeast and mammalian disulfide-biology should not be randomly mixed. And if so, the rational should be declared. Mammalian pathways for the recharge of PDIs with disulfide bonds must therefore be added to the review, preferably in a separate section. This may also include a small text on the issues of oxygen reduction by oxidases (since the authors make the connection to oxidative stress pathways in their conclusions).

Glutathione: I find it impossible to imagine a review on ER-associated disulfide-bond formation without a single mention of glutathione. It is true though that the centrality of glutathione has been challenged by work mainly from the Ron-lab. Thus, compared to earlier reviews on the topic (including the authors' own ones) the paradigm seems to have changed. But if the authors are convinced that the role of glutathione in disulfide-bond formation/rearrangement can be neglected (which I am not), they should prominently state and discuss this (preferably also in their abstract). Completely omitting the documented ER glutathione concentration of several millimolar, which is dynamically regulated by physiological cues, is a bad option though.

pH/pKa: When discussing thiol-disulfide exchange (section 4.1), the role of the surrounding microenvironment is briefly stated. The review would profit from some elaboration. E.g. the role of pH, which directly influences thiol deprotonation and reactivity. The discussion of pH and thiol-pKa is also relevant for section 3.2 where the commonly practiced quenching of thiol-disulfide exchange by rapid acidification is currently missing.

Apart from this, I only have these minor suggestions from improvement:

  • Title: I suggest to replace disulfide formation with "disulfide-bond formation" and use this more precise terminology throughout the text
  • Abstract: Add comma after "in eukaryotic cells" and add analogous commas throughout the text
  • Introduction, para 1: Use either "approximately" or ~, not both
  • Fig1B: For me, the label "folding coupled disulfide formation" refers to the blue pathway only (not the red one). This does not correspond with the current way of labeling.
  • Section 2, last para: BPTI-like and hirudin-like are highly cryptic to non-expert readers and require further explanation
  • Section 3.1: 1 – 6 amino acids per second requires ref.
  • Section 3.1: "extruded into the ER lumen" -> linguistic problem
  • Section 3.1, line 129: add ribosomal before release
  • Fig2: in lower right corner, I would label "Other ER folding factors", because PDIs and chaperones are also ER folding factors
  • Section 3.3, last para: I would add a ref to Fig1B to the b2M, prolactin, disintegrin discussion
  • Section 4.1, last sentence: replace "reductase or oxidase" by "reductant or oxidant". At least the latter is wrong terminology, since an oxidase is defined as an enzyme that reduces molecular oxygen.

Author Response

Reviewer 1

ER oxidases

References on line 35 replaced with reviews by Hudson et al. (2011) and Bulleid and Ellgaard (2011). These are relevant reviews concerning oxidative folding pathways, which are better references for this statement.

As instructed we have added a separate section concerning the pathways for oxidising PDIs (section 4.2), which also includes reference to oxidative stress.

Glutathione

The role of glutathione in buffering redox capacity has been added to the introduction.

pH/pKa

In section 2, lines 83-85 and 103-104 we have added statements concerning the pKa of the thiol group to emphasise its importance early on.

Reference to acid quenching has been added to section 3.2.

The influence of pH on thiol-disulfide exchange was also added to Section 4.1.

Minor corrections

Disulfide changed to “disulfide bond” in the title, abstract, headers and concluding statements. We would prefer to retain “disulfide” throughout the rest of the manuscript.

Commas added where appropriate throughout text.

“~” removed from introduction

We would prefer to keep the labelling in Fig.1B as “folding coupled disulfide formation” as this has been used to describe both folding mechanisms in the past and we think it is the best way to describe the stage between pre-folding and post-folding.

Better explanations of BPTI-like and hirudin-like terminologies have been added to section 2.

Reference added to section 3.1 for the speed of translation (Zhang and Ignatova, 2011). This has also been corrected to 1-5 amino acids per second.

In section 3.1 “extruded” changed to “enter” and “ribosomal” also added where requested.

Labelling on Fig.2 changed to “other ER folding factors”.

In section 3.3 reference to Fig 1B has been added.

In section 4.1 “reductase or oxidase” changed to “reductant or oxidant” as suggested.

Reviewer 2 Report

This review gives readers an overview of the mechanisms of disulfide formation during folding in the ER and is able to provide certain insights into the fields of protein folding. I believe this review is suitable for publication in Cells.

MINOR

(i) Figure 2. What is the black chain on the ribosome? Please add a word (mRNA?) on the scheme.

Author Response

Reviewer 2

mRNA label added to figure 2.

Reviewer 3 Report

Title: Mechanisms of disulfide formation in nascent polypeptides entering the secretory pathway

Authors: Philip Robinson, Neil J. Bulleid * Submitted to section: Organelle Function,

This is a very nicely written review discussing the disulfide formation in the ER and the roles of PDI family proteins. The authors summarized the current knowledge regarding folding and disulfide formation obtained both in vitro and in the cells. Based on the observations in literature, they proposed possible models to understand the complicated process in the ER. The authors also highlighted the emerging problems needing further research. Overall, this review is nicely organized and well-written. The style of the review will facilitate the reading of both new comers and experts of this field.

Minor questions:

Line 58, the authors stated that “Cysteine residues are often buried in the native fold of a protein”. I am not sure about this statement. Disulfides are often in the interior of a protein, but I am not sure whether most cysteines are also buried inside or not. Reference(s) supporting this statement is expected to be cited.

Is there any difference for the co-translational folding and disulfide formation between single-disulfide containing proteins and multi-disulfide containing proteins?

It is better to move the Acknowledgments section to the legend of one related figure such as Figure 3.

Author Response

Reviewer 3

First line in section 2, “Cysteine residues are often buried…..”, has been changed to remove the word buried, and instead we refer to the steric hindrance that occurs when cysteines are folded into the protein. The wording of a similar statement in section 4.3 has also been changed.

There are no known differences between co-translational folding and disulfide formation between single and multi-disulfide containing proteins other than the possibility of non-native disulfide bonds, which we describe in section 4.4. We have added “ for proteins that contain more than 2 cysteine residues” to this section to emphasise this.

Acknowledgements regarding the Chimera software moved to the Figure 3 legend.